# Increased Expression of N2BA Titin Corresponds to More Compliant Myofibrils in Athlete’s Heart

**DOI:** 10.3390/ijms222011110

**Published:** 2021-10-15

**Authors:** Dalma Kellermayer, Bálint Kiss, Hedvig Tordai, Attila Oláh, Henk L. Granzier, Béla Merkely, Miklós Kellermayer, Tamás Radovits

**Affiliations:** 1Heart and Vascular Center, Semmelweis University, Városmajor Str. 68., HU-1122 Budapest, Hungary; dalmakeller@gmail.com (D.K.); olah.attila@med.semmelweis-univ.hu (A.O.); merkely.bela@med.semmelweis-univ.hu (B.M.); 2Department of Biophysics and Radiation Biology, Semmelweis University, Tűzoltó Str. 37–47, HU-1094 Budapest, Hungary; kiss.balint@med.semmelweis-univ.hu (B.K.); tordaih@hegelab.org (H.T.); 3Department of Cellular and Molecular Medicine, University of Arizona, 1656 E Mabel Str., Tucson, AZ 85724-5217, USA; granzier@email.arizona.edu

**Keywords:** titin, athlete’s heart, cardiac myofibril, AFM, transverse stiffness

## Abstract

Long-term exercise induces physiological cardiac adaptation, a condition referred to as athlete’s heart. Exercise tolerance is known to be associated with decreased cardiac passive stiffness. Passive stiffness of the heart muscle is determined by the giant elastic protein titin. The adult cardiac muscle contains two titin isoforms: the more compliant N2BA and the stiffer N2B. Titin-based passive stiffness may be controlled by altering the expression of the different isoforms or via post-translational modifications such as phosphorylation. Currently, there is very limited knowledge about titin’s role in cardiac adaptation during long-term exercise. Our aim was to determine the N2BA/N2B ratio and post-translational phosphorylation of titin in the left ventricle and to correlate the changes with the structure and transverse stiffness of cardiac sarcomeres in a rat model of an athlete’s heart. The athlete’s heart was induced by a 12-week-long swim-based training. In the exercised myocardium the N2BA/N2B ratio was significantly increased, Ser11878 of the PEVK domain was hypophosphorlyated, and the sarcomeric transverse elastic modulus was reduced. Thus, the reduced passive stiffness in the athlete’s heart is likely caused by a shift towards the expression of the longer cardiac titin isoform and a phosphorylation-induced softening of the PEVK domain which is manifested in a mechanical rearrangement locally, within the cardiac sarcomere.

## 1. Introduction

It is widely accepted that regular exercise has a beneficial role in cardiovascular health [1,2]. Long-term physical activity induces complex physiological changes in cardiac morphology and function, referred to as athlete’s heart [1,3,4]. The consequential cardiac remodeling is an important adaptation to exercise, which includes biventricular myocardial hypertrophy and an improved functional and mechanoenergetic status. Previous studies have also focused on the cellular mechanisms of the enhanced myocardial function triggered by exercise [5,6,7]. Functional studies on cardiomyocytes and cardiac trabecular muscles showed greater calcium sensitivity, rate of force production and loaded shortening velocities in exercised rats [5,6,7]. Furthermore, athlete’s heart was shown to be associated with enhanced left ventricular (LV) diastolic function and hence cardiac compliance [1,8,9]. These favorable adaptive features contribute to high cardiac output during endurance training and may prevent age-related adverse cardiac deconditioning [10].

Cardiac compliance is mainly regulated by the giant elastic protein, titin [11]. Titin, encoded by the *ttn* gene, spans half of the sarcomere from the Z-disk to the M-line [12,13]. Its main roles are to determine the elastic, contractile and signaling properties of cardiac and skeletal muscles [14,15]. The I-band region of titin functions as a molecular spring, and it contains constitutively-expressed Ig regions and unique sequences such as the N2B unique sequence and the PEVK element named after its exuberance in proline (P), glutamic acid (E), valine (V) and lysine (K) [11,16]. Titin is alternatively spliced in the I-band region in a process regulated by the RNA Binding Motif 20 protein (RBM20) [17]. This extensive titin mRNA splicing results in different isoform classes. Two main titin isoforms are expressed in the adult cardiac muscle; the longer, more compliant N2BA and the shorter and stiffer N2B titin [12,13,18]. Elevated N2BA/N2B ratio is thus accompanied by reduced passive stiffness and improved ventricular compliance [18,19]. Moreover, the PEVK and N2B elements are known to have specific phosphorylation sites which also allow for post-translational adjustment of titin-based stiffness [20,21,22,23]. Reduced passive stiffness is associated with increased exercise tolerance and cardiac compliance [9,24]. With the exception of a few human studies [1,8], the effects of exercise on titin alterations have been primarily investigated on genetically modified mouse models [16,20,22]. It has been reported that titin’s compliance is modified after acute and chronic exercise via phosphorylation without an effect on titin isoform expression [20,21,22]. By contrast, a recent report by Chung et al. showed increased N2BA titin isoform content in trained rats compared to sedentary rats [24]. It should be stressed that the prior studies used voluntary wheel running that could elicit different types of adaptation than controlled, balanced exercise protocols (e.g., treadmill running, swimming). Previous experiments evaluated passive stiffness changes in skinned LV muscles or cardiomyocytes after exercise. Slater et al. reported reduced passive stiffness of LV wall strips after 3 weeks of voluntary running wheel training [22]. Methawasin et al. showed that when the fraction of longer titin is experimentally increased, mice have higher exercise capacity and cardiomyocytes present reduced passive stiffness [25]. However, no data are available on the impact of titin modifications on cardiac sarcomere structure and mechanics after long-term exercise.

The purpose of this study was to explore whether titin has a role in the improved compliance of athlete’s heart. We used a balanced, well-regulated 12-week-long swimming exercise in rats then examined the titin isoform content and phosphorylation status of the left ventricle. We further sought to investigate whether any exercise-induced titin alterations affect the sarcomeric structure and elasticity of single cardiac myofibrils.

## 2. Results

### 2.1. Induction of Athlete’s Heart

In the present work, we explore the titin-based molecular and sub-cellular mechanisms of stiffness changes associated with long-term physical exercise in a rat experimental model. The body weight of the exercised rats has decreased at the end of the training, but not significantly. Heart weight, heart weight-to-body weight ratio (HW/BW) and heart weight-to-tibia length ratio (HW/TL) were significantly increased in the exercised (Ex) group reflecting cardiac hypertrophy and athlete’s heart (Table 1).

### 2.2. Effect of Long-Term Exercise on Titin Isoform Content

The effect of long-term exercise on titin isoform content was assessed by measuring the N2BA/N2B ratio in the left ventricle. The N2BA/N2B ratio was significantly increased in the exercised rats (Figure 1A–C), therefore titin expression was shifted towards the more compliant isoform. The relative expression of total titin (TT/MHC) (Figure 1D) and T2 (T2/TT) (Figure 1E) were unaltered in the two groups.

### 2.3. Exercise-Induced Phosphorylation Modifications on Titin

Several phosphosites have been identified in the I-band region of titin. Here, we investigated total titin phosphorylation and alterations in titin’s PEVK phosphosites on Ser118787 and Ser12022 residues. No differences were found between the groups in total phosphorylation by Pro-Q Diamond staining (Figure 2A). Western blot revealed no differences in PS12022 (Figure 2C). By contrast, hypophosphorylation of the PS11878 site (Figure 2B) was detected in exercised rats, pointing at an exercise-specific phosphorylation effect.

### 2.4. Myofibril Sarcomere Structure and Elasticity

The topographical structure of single permeabilized cardiac myofibrils revealed slight differences in the sarcomeric structural dimensions (Figure 3A) (*n* = 63 Co vs. *n* = 52 Ex sarcomeres). The sarcomere length (SL) was measured as the distance between the neighboring Z-disks on the surface profile of each myofibril [26]. The length between the two deepest points surrounding the Z-disk was determined as the I-band length, while the length between the two deepest points surrounding the M-band corresponds to the A-band length (Figure 3B) [26]. To account for skew, each sarcomere was measured along 3 paraxial contour profiles; one in the center and one on each side. The sarcomeric structure was well recognizable on the myofibrils. We note that the Ex myofibrils displayed greater flexibility which manifested in an overall more curved structure, while Co myofibrils had a more straight appearance, which points at a greater bending rigidity. Notably, Co myofibrils retained a more regular sarcomeric structure while the Ex myofibrils displayed a greater level of contour and surface irregularities. The SL was in the normal slack length range (1.7–2.2 µm) in both groups [27], although we measured significantly shorter sarcomere lengths in the Ex group (Co = 1.81 ± 0.01 µm vs. Ex = 1.73 ± 0.02 µm, *p* < 0.01) (Figure 3C). The muscles were immediately flash frozen in liquid nitrogen following the dissection, and no stretching was applied on the tissue once it was placed in the permeabilizing solution. Although the myofibrils were exposed to a buffer environment mimicking rigor conditions, internal stores of ATP may have contributed to a small degree of sarcomeric contraction during sample preparation. The I-band length normalized to the SL and the A-band length normalized to the SL did not differ in the two groups (Figure 3D). The I-band/Z-disk height and the I-band/M-band height did not show alterations between the Co and Ex groups (Figure 3E). The Z-disk/M-band height was significantly decreased in Ex myofibrils (Figure 3F). The Z-disk region is not only an important anchoring site for sarcomeric proteins, but it has additional roles in intracellular signaling, mechanosensation and protein turnover.

Fast force mapping (FFM) was performed to examine the lateral stiffness of Co and Ex sarcomeres, where each pixel gives a force curve (Figure 4A). The Young’s modulus was calculated with the Johnson–Kendall–Roberts (JKR) fitting model of the force curves. Myofibrils showed a significantly decreased Young’s modulus after long-term exercise (Co = 3.56 ± 1.17 MPa vs. Ex = 1.35 ± 1.51 MPa, *p* < 0.01) (Figure 4B). Conceivably, the irregular surface and contour of Ex myofibrils are caused by the increased sarcomeric compliance.

## 3. Discussion

An athlete’s heart adapts to regular exercise and results in morphological and functional changes of the cardiovascular system [1,2,3]. The importance of this phenomenon is underlined by the increasing number of professional athletes and the demand for physical fitness in the general population [28]. It is differentiated from cardiomyopathies and is considered to be a reversible, physiological condition of the heart [28,29]. It has been extensively documented that aerobic exercise induces atrial and ventricular hypertrophy [1,3,4,30]. These adaptations lead to increased cardiac output, stroke volume, improved contractility, relaxation and mechanoenergetic status [4,5,6,7]. The endurance athlete’s heart is also associated with supernormal left ventricular (LV) cardiac compliance [1,8,9]. Cardiac compliance is primarily regulated by titin, the giant elastic myofilament protein [11]. However, modifications of titin induced by extensive exercise and its potential contribution to the improved performance of an athlete’s heart have not been studied thoroughly. To our knowledge, this study is the first to evaluate titin isoform expression and phosphorylation of the athlete’s heart and its impact on myofibrils.

### 3.1. Athlete’s Heart Induced by 12-Week-Long Swimming Training

Several methods have been established to evaluate LV hypertrophy and thus athlete’s heart by exercise [31]. We induced athlete’s heart by a 12-week-long swimming-based training, according to our previously published and well-established protocol [4]. Swimming exercise provides balanced endurance training. It is well controlled and regulated, as we intended to minimize the differences in exercise duration and intensity between the rats. We have previously evaluated the morphological and functional parameters of an athlete’s heart comprehensively by using swimming training [4,32,33,34]. In the current study, control rats also swam 5 min/day to eliminate the stress effects caused by water [4]. Exercised rats showed significantly increased heart weight, HW/BW and HW/TL, indicating that physiological hypertrophy has indeed developed (Table 1.).

### 3.2. Exercise-Induced Titin Isoform Expression Alteration

Titin is the major determinant of passive stiffness in striated muscle [11,14]. It has been well documented that titin-based passive stiffness predicts exercise tolerance [9]. Titin is a target for alternative splicing by the RNA Binding Motif 20 protein (RBM20) [17], resulting in two main adult cardiac isoforms: the shorter, stiffer N2B and the longer, more compliant N2BA [12,13]. The ratio of these isoforms defines the overall passive stiffness in cardiac muscle. Elevated N2BA content, and thus an increased N2BA/N2B ratio, is accompanied by reduced passive stiffness and corresponds to a more compliant heart [18,19]. Nagueh et al. also suggested that the greater N2BA/N2BA ratio in patients with dilated cardiomyopathy is associated with improved exercise tolerance [19].

Here, we showed considerable alterations in titin isoform expression by using the rat model of the athlete’s heart. We found a significantly increased N2BA/N2B isoform ratio in the exercised group, pointing to a more compliant heart after training (Figure 1). No changes were detected in total titin or T2 content in the two groups. Previous studies using 3–6 weeks voluntary running wheel in mouse models did not reveal any titin isoform changes [20,22]. Recently, Chung et al. showed an increased N2BA/N2B ratio in trained versus sedentary rats [24]. In their study, rats had access to a voluntary running wheel for 12 weeks. Although voluntary running wheel exercise is an acceptable model to investigate left ventricular hypertrophy, our swimming program is more regulated and uniform and mimics the exercise protocol of professional athletes. Our titin data are in agreement with the findings of Chung et al.; however, in our study, a smaller sample size was sufficient to detect differences in titin isoform content.

We propose that long-term, balanced and regulated training is sufficient to demonstrate N2BA/N2B ratio alterations with a lower sample number. More importantly, the upregulation of the longer N2BA titin appears to be an adaptive mechanism in response to the increased diastolic demand in the athlete’s heart.

### 3.3. Post-Translational Modifications of Titin after Long-Term Exercise

Titin-based passive stiffness can be modulated by post-translational modifications, of which the most extensively evaluated process is phosphorylation [20,21,22,35]. Titin has multiple phosphorylation sites [23] with a few outstanding, well-described specific phosphosites in the N2B unique sequence (N2Bus) and in the PEVK element. In the N2Bus, Ser4010 (targeted by PKA and ERK1/2) and Ser4099 (targeted by PKG) motifs have been characterized [20,21]. The most investigated phosphosites of the PEVK element are Ser11878 and Ser12022 residues (targeted by PKCα and CaMKIIδ) [21,35]. It has been previously shown that titin-based passive stiffness is reduced by the phosphorylation of the N2Bus [36], whereas phosphorylation of the PEVK region increases stiffness [22,35]. In our study, we revealed no changes in total titin phosphorylation, similar to the findings of Chung et al. (Figure 2) [24]. However, our analysis showed significantly decreased phosphorylation of the Ser11878 site in athlete’s heart, while the phosphorylation of the Ser12022 remained unchanged, suggesting that passive stiffness of the heart after long-term exercise becomes reduced via a PEVK phosphorylation mechanism. This is in agreement with the findings of Slater et al., who additionally detected hyperphosphorylation of the Ser4010 in the N2Bus [22]. By contrast, hypophosphorylation of the Ser12022 and unaltered phosphorylation of the Ser11878 residue has been reported by Hidalgo et al. after chronic exercise [20]. The difference in the phosphorylation spectrum is possibly due to the distinct binding of PKCα to the PEVK element, with a greater affinity to Ser11878 [37]. A previous study by Muller et al. also investigated titin’s post-translational phosphorylation evoked by short-term exercise [21]. In total, 15 min of treadmill running induced a decrease in the phosphorylation of the Ser4099 residue and hyperphosphorylation of the Ser11878 site in LV samples, leading to increased titin-based stiffness. The increased stiffness may contribute to the Frank–Starling mechanism and improve cardiac output to adapt to the rapid volume changes during acute exercise [21,25].

Thus, our findings predict that chronic exercise leads to reduced titin-based passive stiffness in the athlete’s heart by hypophosphorylation in the PEVK element.

### 3.4. Structure and Elasticity of Exercise-Exposed Myofibrils

To test the prediction that chronic exercise might lead to a reduction of sarcomeric stiffness, we measured the transverse elasticity of single myofibrils isolated from the left ventricles of rats exposed to chronic exercise. We used atomic force microscopy (AFM) to image and mechanically manipulate single, substrate-attached cardiac myofibrils under buffer solution conditions. The myofibrils underwent a fast centrifuge step in order to enhance the fixation to the mica surface and this affected the exercised myofibrils to a higher degree. By using resonant (non-contact or AC mode) scanning, we observed well recognizable sarcomere structures in both groups. However, the Ex myofibrils showed more flexible and bent sarcomeres, while control myofibrils displayed straighter sarcomeres in-series. Thus, the Ex myofibrils presented a greater level of contour and surface irregularities, which may have been induced by the mechanical effects of the exercise. We demonstrated sarcomere lengths (SL) of the in vivo working length in both groups (1.7–2.2 µm) [27]. Nevertheless, we observed shorter sarcomeres in the Ex myofibrils (Co = 1.81 ± 0.01 µm vs. Ex = 1.73 ± 0.02 µm, *p* < 0.01). The LV samples were immediately frozen in liquid nitrogen after dissection and no stretching was applied during the myofibril preparation. We added 2,3-butanedione-monoxime (BDM) in the permeabilization solution to prevent further contraction of the samples, therefore the myofibrils possibly kept their initial length during freezing. However, it cannot be ruled out that internal stores of ATP may have contributed to a small degree of sarcomeric contraction. Titin’s I-band region acts as a molecular spring [11]. A larger relative amount of the more compliant N2BA titin would allow for larger-amplitude contractions and could also underlie the shorter sarcomeres in the Ex myofibrils. We used the method of Ogneva et al. to measure the I-, and A-band-lengths as this was more reproducible than measuring the distances from the half height of the Z-disk and M-band [26]. The I- and A-band length normalized to SL did not show any differences in the two groups. No alterations were found in the I-band height/Z-disk height and the I-band height/M-band height. Interestingly, the Z-disk height/M-band height was decreased in the Ex myofibrils. It is possible that the Ex myofibrils became more flattened after the centrifuge step, contributing to the irregularity of the Ex myofibrils. Furthermore, the Young’s modulus (elastic modulus) of the myofibrils, determined by fast force mapping (FFM) mode of the AFM, was decreased in the exercised group, providing further evidence that passive stiffness is reduced in trained myofibrils. Although the titin-specific isoform and phosphorylation changes are spatially confined in the rather short I-band, interestingly, the mechanical changes are manifested across the entire sarcomere. Further research may reveal the molecular mechanisms behind this observation.

Few groups investigated the Young’s modulus of striated muscle by AFM. Li et al. examined the lateral stiffness of psoas and diaphragm muscles and concluded that titin contributes to the A-band lateral stiffness and, thereby, is involved in active contraction [38]. A study by Akiyama et al. evaluated control cardiac myofibrils and provided lower Young’s moduli compared to our results [39]. However, they used neonatal LV muscle, which is known to express fetal N2BA, the most compliant titin isoform [18]. Nevertheless, no prior studies were performed to evaluate the Young’s modulus of control or trained adult cardiac myofibrils by FFM atomic force microscopy.

In sum, our results indicate that chronic exercise leads to an increase in sarcomeric compliance. We speculate that this change in sarcomere mechanics, which is closely linked to a shift in the titin isoform ratio and PEVK phosphorylation spectrum, is part of an important response mechanism that aids the physiological adaptation to the demands of physical exercise.

## 4. Materials and Methods

### 4.1. Animal Model and Ethical Approval

All experimental procedures were approved by the Ethical Committee of Hungary for Animal Experimentation in accordance with the ‘‘Principles of Laboratory Animal Care’’ defined by the National Society for Medical Research and the Guide for the Care and Use of Laboratory Animals, provided by the Institute of Laboratory Animal Resources and published by the National Institutes of Health (publication no. 85-23, revised 1996) and the European Union Directive 2010/63/EU. Rat tissue samples were collected by using an established protocol of exercise-induced cardiac hypertrophy by swimming, as previously described [4]. Briefly, after acclimatization, young adult Wistar male rats (12-week-old, *n* = 12, m= 275–325 g) (Charles River, Sulzfeld, Germany) were randomly divided into control (Co, *n* = 6) and exercised (Ex, *n* = 6) groups. Swimming exercise was performed in a divided container filled with tap water (45 cm deep) maintained at 30–32 °C. Rats of the exercised group were exposed to 200 min/day of swimming 5 days/week for 12 weeks to induce physiological LV hypertrophy, thereby athlete’s heart. In order to reach an appropriate level of adaptation, the duration of swimming was increased by 15 min every second training day after a basic 15 min on the first day, until the maximal 200 min/day was reached. During the 12-wk-long training period, control animals were habituated to water 5 min/day to abolish the possible differences induced by the stress of water contact. All rats were housed in standard rat cages at a constant room temperature (22 ± 2 °C) with a 12-h light–12-h dark cycle. Animals were fed with standard rodent chow and water ad libitum. After 12 weeks of exercise, the rats were euthanized, following a 24-h rest period, by exsanguination under deep anaesthesia. The weight of the excised heart was measured immediately. The heart was dissected and the LV was flash frozen in liquid nitrogen and subsequently stored at −80 °C.

### 4.2. Sample Solubilization

Small segments (~10–15 mg) of LV muscles were homogenized using glass Kontes Dounce tissue grinders cooled in liquid nitrogen [18]. Tissues were primed at −20 °C for a minimum of 20 min, then solubilized in 50% urea buffer ([in mol/L] 8 Urea, 2 Thiourea, 0.05 Tris-HCl, 0.075 Dithiothreitol with 3% SDS and 0.03% Bromophenol blue, pH 6.8) and 50% glycerol with protease inhibitors ([in mmol/L] 0.04 E64, 0.16 Leupeptin and 0.2 PMSF) at 60 °C. Solubilized samples were centrifuged for 5 min, aliquoted and flash frozen in liquid nitrogen and stored at −80 °C.

### 4.3. Titin Isoform Analysis and Total Titin Phosphorylation

Titin isoform expression was determined by large format 1% sodium-dodecyl-sulfate (SDS)—agarose gel electrophoresis [40]. Gel electrophoresis was performed at 16 mA/gel for 3 h 30 min. Gels were stained with SYPRO Ruby Protein Gel Stain (Thermo Fischer Scientific, Waltham, MA, USA), which gives the maximum signal strength and widest linear dynamic range for protein quantification [41]. Gels were scanned by using a Typhoon laser-scanner (Amersham BioSciences, Little Chalfont, Buckinghamshire, United Kingdom), and optical density was analyzed with ImageJ software (ImageJ 1.52k, National Institutes of Health, Bethesda, MD, USA). Titin isoform ratio (N2BA/N2B) was calculated from the integrated band densities. Relative content of total titin (TT) was normalized to myosin heavy chain (MHC), and T2 (titin’s proteolytic degradation product) was normalized to TT. The two N2BA bands were summed to determine the relative N2BA isoform content [24,42]. To determine total titin phosphorylation, 1% SDS-agarose gel electrophoresis was performed. Gels were stained for phosphoprotein using Pro-Q Diamond staining (Thermo Fisher Scientific, Waltham, MA, USA) and scanned with a Typhoon laser scanner. Gels were then stained for total protein content with SYPRO Ruby Protein Gel Stain and scanned with a Typhoon laser scanner. Bands were quantified with the ImageJ software.

### 4.4. Titin’s PEVK Site Specific Phosphorylation

To determine the phosphorylation level of titin’s PEVK phosphosites, samples were first separated on a 0.8% SDS-agarose gel. The separated proteins were then transferred onto PVDF membranes (Hybond-LFP, Amersham BioSciences, Little Chalfont, Buckinghamshire, United Kingdom) with a semi-dry blotter unit (Trans-Blot Cell, Bio-Rad, Hercules, CA, USA). Subsequently, the blots were probed with phospho-specific anti-pS11878 and pS12022 [20,22] and anti-T12 antibody (detecting titin’s N-terminus, kindly provided by Dieter O. Fürst, University of Bonn, Germany) [43] primary antibodies overnight at 4 °C, followed by secondary CyDye conjugated antibodies (Amersham BioSciences, Little Chalfont, Buckinghamshire, United Kingdom). Blots were scanned with a Typhoon laser scanner. The phosphorylation signal was normalized to the T12 antibody signal. Relative expressions of the proteins were analyzed with ImageJ.

### 4.5. Preparation of Single Cardiac Myofibrils

The preparation of myofibrils was based on previously described methods [44,45]. Briefly, 1–3 small pieces of LV muscle samples (total weight of 7–10 mg), frozen in liquid nitrogen, were immersed and quickly added to a 2ml Eppendorf tube filled with ice-cold permeabilization solution ([in mmol/L] 10 Tris (pH 7.1), 132 NaCl, 5 KCl, 1 MgCl_2_, 5 EGTA, 5 dithiothreitol (DTT), 10 NaN_3_, 20 2,3-butanedione-monoxime (BDM), 1% Triton X-100) with protease inhibitors ([in mmol/L] 0.04 E64, 0.16 Leupeptin and 0.2 PMSF). BDM prevented contraction and increased storage time. The samples were placed on a 360° rotating shaker for 3 h at 4 °C. After permeabilization, the solution was removed and the permeabilized samples were washed in washing solution (as permeabilization solution, without Triton X-100 and BDM) for 15min. To prepare a suspension of myofibrils, the samples were transferred into 1mL ice-cold myofibril buffer ([in mmol/L] 100 KCl, 2 MgCl_2_, 1 EGTA, 0.5 DTT, 10 KH_2_PO_4_, pH 7.1) [44] and homogenized with an MT-30K Handheld Homogenizer at 27,000 rpm for 10–15 s (Hangzhou Miu Instruments Co. Ltd., Hangzhou, Zhejiang, China). The myofibrils were pelleted by low-speed centrifugation, the pellet was re-suspended in 300 µL of fresh myofibril buffer and kept on ice until use for experiments for up to 2 days [44,45].

### 4.6. Atomic Force Microscopy (AFM) Imaging and Force Spectroscopy

To prepare an appropriate surface for myofibrils, 100 µL of 0.1% *w*/*v* poly-L-lysine (PLL) (Merck, Darmstadt, Germany) solution was pipetted onto a freshly cleaved mica surface (Ted Pella, Redding, CA, USA) and incubated for 20 min. Then, the surface was rinsed with distilled water and dried in high purity nitrogen gas (N_2_) stream. Subsequently, 100 µL of 25% *w/v* grade I glutaraldehyde (GA) (Merck, Darmstadt, Germany) was pipetted on the surface and incubated for 20 min, followed by further rinsing with distilled water and drying in N_2_ stream. An aliquot (100 µL) of the myofibril sample was pipetted onto the previously prepared mica surface and was incubated for 15 min. After incubation, the surface was centrifuged at 13,000 rpm for 1 s to enhance the fixation of the myofibrils, followed by immediate rehydration with myofibril buffer. Subsequently, the surface was rinsed gently with myofibril buffer to remove unbound myofibrils. To examine sarcomere structure and elasticity of single, relaxed myofibrils, AFM imaging was carried out with an Asylum Research Cypher ES instrument (Oxford Instruments, Santa Barbara, CA, USA) [46]. Resonant-mode (AC- or non-contact mode) scanning was performed under liquid (myofibril buffer) with silicon nitride cantilevers (BL-AC40TS, Olympus Corporation, Shinjuku, Tokyo, Japan). The cantilever was oscillated near its resonance frequency (typically around 20 kHz) by using photothermal excitation (BlueDrive technology) at a typical free amplitude of 0.5 V. Prior to imaging, cantilever spring constants were determined by the thermal method [47]. Spring constants were between 90–120 pN/nm. Imaging was carried out at a typical setpoint of 350 mV at room temperature. The AC mode image was used to determine sarcomere length and to identify the Z-disk, I-band, A-band and M-band regions on the surface profile of each myofibril. The neighboring Z-disks determined the sarcomere length. We calculated the I-band length as the length of the two deepest points surrounding the Z-disk, while the length between the two deepest points surrounding the M-band determined the A-band length [26]. For force spectroscopic and transverse elasticity measurements we used fast force mapping (FFM, or “jumping-mode” AFM) [48]. In FFM, the cantilever was driven sinusoidally with a typical frequency of 300 Hz and a setpoint force of 500 pN to obtain a force curve for each pixel [49]. The spatially resolved Young-modulus map of the sarcomere, which corresponds to elasticity against compressive forces, was obtained with the Johnson–Kendall–Roberts (JKR) fitting model of the force curves [50].

### 4.7. Data Analysis and Statistics

Data are presented as mean ± SEM. Statistics were calculated with GraphPad Prism (GraphPad Software, La Jolla, CA, USA). Normal distribution was tested using the Shapiro–Wilk test and unpaired Student’s t-test was performed. *p*-value < 0.05 was considered to be statistically significant. For AFM, image postprocessing and data analysis were performed by using the AFM driving software AR16, IgorPro 6.37 (Wavemetrics, Lake Oswego, OR, USA).

## 5. Conclusions

In summary, we investigated titin alterations under physiological conditions that mimic the circumstances of highly trained professional athletes. The increased N2BA/N2B ratio associated with the hypophosphorylation of the PEVK element indicates decreased titin-based passive stiffness and a more compliant heart after long-term exercise. The increased expression of N2BA potentially led to more flexible myofibrils in the exercised group. The decreased Young’s modulus in the exercised group indicates reduced stiffness and more compliant myofibrils following chronic exercise. Therefore, titin modifications after long-term exercise are potential adaptive mechanisms that contribute to an improved cardiac compliance and favorable performance of the athlete’s heart.

## Figures and Tables

**Figure 1 ijms-22-11110-f001:**
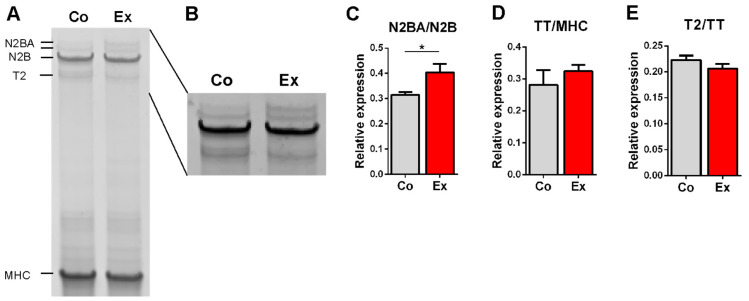
Titin isoform analysis. (**A**) Example gel electrophoresis of a control and exercised LV sample. (**B**) Linear contrast adjustment also applied to the original image for better visualization. (**C**) The ratio of the more compliant N2BA and the stiffer N2B titin isoform is increased in exercised rats. (**D**) Total titin (TT) to Myosin Heavy Chain (MHC) ratio and (**E**) the titin degradation product T2 relative to TT did not differ between the two groups. *n* = 6/group, * *p* < 0.05.

**Figure 2 ijms-22-11110-f002:**
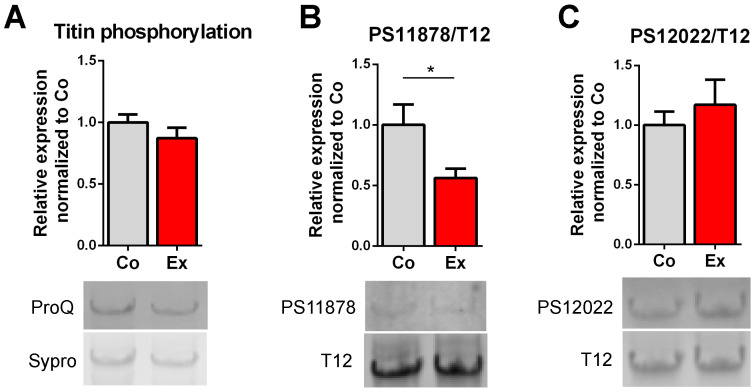
Effect of long-term exercise on titin phosphorylation. (**A**) Total titin phosphorylation did not differ between control and exercised rats. (**B**) Exercise caused hypophosphorylation of titin’s S11878 (S26) site (linear contrast adjustment was applied to the original PS11878 image for better visualization), (**C**) but had no effect on S12022 (S170). *n* = 6/group, * *p* < 0.05. Sypro: Sypro Ruby Protein Gel Stain, T12: anti-12 antibody detecting titin’s N-terminus.

**Figure 3 ijms-22-11110-f003:**
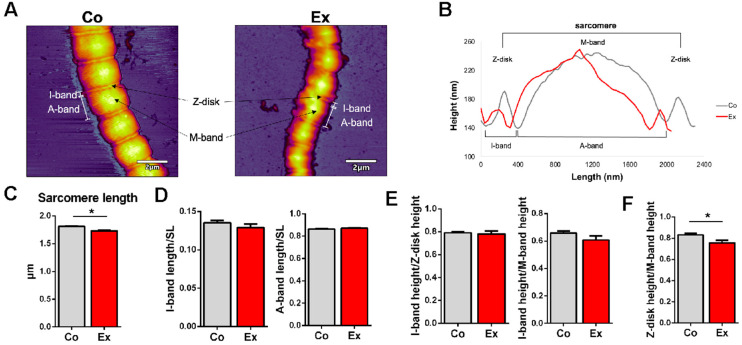
Topographical structure of relaxed control and exercised myofibrils with Atomic Force Microscopy (AFM)**.** (**A**) Representative AFM images of a Co and Ex myofibril demonstrating the sarcomere structure. The Ex myofibril displays more flexible and bent sarcomeres. Scale bar = 2 μm. (**B**) Representative topographical surface profile of a Co and Ex sample. (**C**) The sarcomere length (SL) was shorter in the exercised group. (**D**) The I-band length/SL and A-band length/SL was unaltered in the two groups. (**E**) There were no differences in the I-band height/Z-disk height and A-band height/Z-disk height. (**F**) The Z-disk height/M-band height was decreased in the Ex group. *n* = 63 Co sarcomeres (4 hearts) vs. *n* = 52 Ex sarcomeres (3 hearts), * *p* < 0.05.

**Figure 4 ijms-22-11110-f004:**
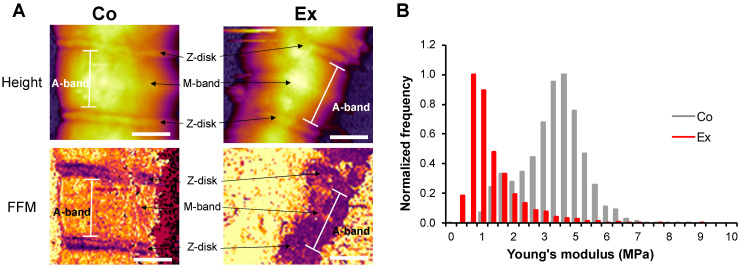
Fast force mapping of sarcomeres with AFM. (**A**) Height image and stiffness map of a Co and Ex sarcomere at a trigger force of 500 pN. (**B**) The Young’s modulus is decreased in the Ex group vs. Co. Young’s modulus was calculated with the Johnson-Kendall-Roberts (JKR) fitting model. Note that the Ex sarcomere showed more perturbation. Scale bar = 1 µm.

**Table 1 ijms-22-11110-t001:** Body and heart weight parameters.

	Co (*n* = 6)	Ex (*n* = 6)	*p*-Value
BW (g)	483 ± 24	417 ± 18	0.06
TL (cm)	4.33 ± 0.05	4.18 ± 0.06	0.09
HW (g)	1.23 ± 0.05	1.45 ± 0.08 *	0.04
HW/BW (g/kg)	2.55 ± 0.08	3.47 ± 0.09 *	<0.01
HW/TL (g/cm)	0.28 ± 0.01	0.34 ± 0.01 *	<0.01

The exercised group shows significantly increased heart weight data. Data are expressed as mean ± SEM. Co: control; Ex: exercised; BW: body weight, TL: tibia length, HW: heart weight, HW/BW: heart weight-to-body weight ratio, HW/TL: heart weight-to-tibia length ratio. * *p* < 0.05

## Data Availability

The data presented in this study are available on request from the corresponding author.

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
