# Peer review of "Increased Expression of N2BA Titin Corresponds to More Compliant Myofibrils in Athlete’s Heart"

_ijms, 2021, doi:10.3390/ijms222011110_

Round 1

Reviewer 1 Report

General

The manuscript entitled “Increased expression of N2BA titin corresponds to more compliant myofibrils in athlete’s heart” by Kellermayer, et al is seeks to delineate the role of titin isoform expression changes in regulation of cardiac passive tension in exercised rats.  The manuscript is well written and the data largely support the authors’ conclusions. However, the authors’ key findings have been previously described and the new data presented in this manuscript represent only an incremental increase in understanding titin regulation of passive tension in exercised hearts. The authors address this concern by stating that they used a more rigorous, controlled exercise program, but the data reported are completely consistent with all other models of exercise in which titin expression was examined.  The unique finding of S11878 hypophosphorylation is consistent with literature demonstrating that hyperphosphorylation of S11878 is associated with increases in passive stiffness. This is the first report of S11878 hypophosphorylation in exercise and concomitant reduction in passive stiffeness. However, the blot shown in Fig 2B is not convincing. Presumably this is the best blot that the authors have generated and it appears that the authors are working right at the limit of detection of the antibody (the same concern applies to the PQD image shown in 2A). Lastly, the myofibril data, while consistent with published data (of decreased passive tension in the exercised heart) reveal a more disordered sarcomere. Of more concern, the sarcomere lengths reported are somewhat short. The authors attempt to discuss the potential mechanisms (both physiological and experimental), but it raises some concerns with the fidelity of the data. It would be important to demonstrate in a larger preparation (skinned cell, skinned fiber, etc) that changes in tension were maintained in the whole cell/tissue.

Specific concerns:

  1. P3, line 108 is a fragment
  2. The quality of the signals shown in the gels and blots shown in Fig2 is insufficient.
  3. P5, line 148; the last word “end” should be “and”
  4. P8, lines 304-306. It is unclear whether training was discontinued for 24-48 hours before euthanasia (in order to minimize and acute exercise mediated changes in titin phosphorylation). Please state the temporal association between the last exercise training and euthanasia.

Author Response

Response to Reviewer 1

We thank the reviewer for the kind and encouraging critical comments which helped improving our manuscript further. Please, find below our detailed response to the comments and questions.

 Comment:

The manuscript entitled “Increased expression of N2BA titin corresponds to more compliant myofibrils in athlete’s heart” by Kellermayer, et al is seeks to delineate the role of titin isoform expression changes in regulation of cardiac passive tension in exercised rats. The manuscript is well written and the data largely support the authors’ conclusions. However, the authors’ key findings have been previously described and the new data presented in this manuscript represent only an incremental increase in understanding titin regulation of passive tension in exercised hearts. The authors address this concern by stating that they used a more rigorous, controlled exercise program, but the data reported are completely consistent with all other models of exercise in which titin expression was examined.

Response:

While the titin-associated sarcomeric changes occurring in the exercised cardiac muscle have indeed been investigated and documented before, we are convinced that the present work carries novelty by the type of the exercise involved and the AFM-based nanomechanical investigations. In this regard the results strengthen the notion that titin plays a fundamental role in the structural and mechanical control of sarcomeric function via alterations in mRNA splicing and localized protein phosphorylation.

Comment:

The unique finding of S11878 hypophosphorylation is consistent with literature demonstrating that hyperphosphorylation of S11878 is associated with increases in passive stiffness. This is the first report of S11878 hypophosphorylation in exercise and concomitant reduction in passive stiffeness. However, the blot shown in Fig 2B is not convincing. Presumably this is the best blot that the authors have generated and it appears that the authors are working right at the limit of detection of the antibody (the same concern applies to the PQD image shown in 2A).

Response:

Thank you for pointing out the visually somewhat poor western blots. We find that this issue is probably a recurring problem in the analysis of similar experimental data, as this phenomenon has been observed in prior publications (cf. Mohamed et al. Eur. J. Heart Fail. 18, 362, 2016, supplementary material). In the revised manuscript we attached a new version of the ProQ Diamond stained gel, which was recorded at an increased detector sensitivity. We also note that in spite of the low visual quality, densitometric analysis was able to uncover the differences between the control and exercised samples.

Comment:

Lastly, the myofibril data, while consistent with published data (of decreased passive tension in the exercised heart) reveal a more disordered sarcomere. Of more concern, the sarcomere lengths reported are somewhat short. The authors attempt to discuss the potential mechanisms (both physiological and experimental), but it raises some concerns with the fidelity of the data. It would be important to demonstrate in a larger preparation (skinned cell, skinned fiber, etc) that changes in tension were maintained in the whole cell/tissue.

Response:

Thank you for raising the issue of mechanical aspects measured in larger-size samples. We note here that such measurements have been reported before and duly referenced in our manuscript (see Introduction). The shorter-than-usual sarcomeres may be associated with the sample preparation method. The somewhat disordered appearance of the exercised myofibrils may actually be caused by the exercise itself. We added a comment on this possibility in the revised manuscript. Notably, in the present work we intended to uncover the microscopic-scale nanomechanical changes that could be associated with titin's isoform and phosphorylation changes. While the exact molecular mechanisms behind the discovered changes await further exploration, the findings reveal that the titin-associated changes affect the supramolecular structure and mechanics of the entire sarcomere.

Specific concerns:

  1. P3, line 108 is a fragment
    Response:
    Thank you for pointing out the missing words. We have corrected the problem.

  2. The quality of the signals shown in the gels and blots shown in Fig2 is insufficient.
    Response:
    Indeed, the image quality of the gels and blots may appear poor by visual inspection. To aid the evaluation, we attached a new version of the ProQ Diamond stained gel in the revised manuscript, in which the detector sensitivity was increased. Furthermore, we note on one hand that our phosphosite-specific western blots are on par with some results in the relevant literature ( Mohamed et al. Eur. J. Heart Fail. 18, 362, 2016, supplementary material) and, on the other hand, densitogram analysis did reveal differences in spite of the relatively poor visual appearance.

  3. P5, line 148; the last word “end” should be “and”
    Response:
    Thank you for noting the error, which we corrected in the revised manuscript.

  4. P8, lines 304-306. It is unclear whether training was discontinued for 24-48 hours before euthanasia (in order to minimize and acute exercise mediated changes in titin phosphorylation). Please state the temporal association between the last exercise training and euthanasia.
    Response:
    Thank you for pointing at this issue. In fact, the rats were rested for 24 hours after the training and prior to euthanasia. We amended the text accordingly.

Reviewer 2 Report

Dear authors,

I read with pleasure your article and I have only minor concerns:

The use of SEM doesn’t seem to me always appropriate. For example, I believe that the parameters reported in table 1 and the SL’s as reported in lines 129 and 130 would be more appropriately described by the SD. Indeed, for what I understand of statistics, those parameters have intrinsically dispersed values, that report the variability in the population rather than the error in the measurement.

A second concern is about the relation between the passive stiffness associated with either N2BA or N2B and the transverse stiffness of the sarcomeres. The SL’s of the myofibrils are very short. Luther et al 2008, JMB 384:60-72 report a thick filament (or A-band) length of 1.584±0.011 μm, and Luther 2009 J Muscle Res Cell Motil 30:171–185 a Z-disk width of about 0.1 μm in heart. Then, SL = 1.73 μm implies that I-band titin has only about 25 nm available along the axial extension of the half-sarcomere. In this situation of folding, it seems to me unlikely that the different stiffnesses of the two isoforms can unveil their effect on the transverse stiffness. Could you better clarify this point?

Author Response

Response to Reviewer 2

We thank the reviewer for the kind and encouraging comments which helped improving our manuscript further. Please, find below the detailed response to the comments and questions.

Comment:

I read with pleasure your article and I have only minor concerns:

The use of SEM doesn’t seem to me always appropriate. For example, I believe that the parameters reported in table 1 and the SL’s as reported in lines 129 and 130 would be more appropriately described by the SD. Indeed, for what I understand of statistics, those parameters have intrinsically dispersed values, that report the variability in the population rather than the error in the measurement.

Response:

We prefer the use of SEM simply because it incorporates the random variation in the investigated variable besides the statistical variation associated with the measurement itself. Accordingly, we opted for keeping with the use of SEM in the revised manuscript version.

Comment:

A second concern is about the relation between the passive stiffness associated with either N2BA or N2B and the transverse stiffness of the sarcomeres. The SL’s of the myofibrils are very short. Luther et al 2008, JMB 384:60-72 report a thick filament (or A-band) length of 1.584±0.011 μm, and Luther 2009 J Muscle Res Cell Motil 30:171–185 a Z-disk width of about 0.1 μm in heart. Then, SL = 1.73 μm implies that I-band titin has only about 25 nm available along the axial extension of the half-sarcomere. In this situation of folding, it seems to me unlikely that the different stiffnesses of the two isoforms can unveil their effect on the transverse stiffness. Could you better clarify this point?

Response:

Thank you for raising the issue of sarcomere length and the consequential problems of lateral stiffness. The utilized AFM method, which unveils the topographical structure only, is not able to reveal the exact molecular arrangement of titin in the indeed rather small space available at the short sarcomere lengths. However, it is a fact that in the exercised sarcomere the lateral compliance is increased in the case of the increased N2BA/N2B titin isoform ratio at a slightly reduced sarcomere length. Conceivably a yet unknown molecular rearrangement of titin occurs in the I-band of these sarcomeres. In the Discussion of the revised manuscript we added a brief note on this possibility.

Round 2

Reviewer 1 Report

The revised manuscript has address all my previous concerns.